# Long-Term Impacts of Fuel Treatment Placement with Respect to Forest Cover Type on Potential Fire Behavior across a Mountainous Landscape

**Seth A. Ex \*, Justin P. Ziegler, Wade T. Tinkham** 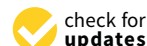 **and Chad M. Hoffman**

Department of Forest and Rangeland Stewardship, Colorado State University, 1472 Campus Delivery, Fort Collins, CO 80523, USA; justin.ziegler@colostate.edu (J.P.Z.); Wade.Tinkham@colostate.edu (W.T.T.); c.hoffman@colostate.edu (C.M.H.)

**\*** Correspondence: seth.ex@colostate.edu; Tel.: +1-970-491-5913

**Abstract:** *Research Highlights:* The impact of variation in fuels and fuel dynamics among forest cover types on the outcome of fuel treatments is poorly understood. This study investigated the potential effects of treatment placement with respect to cover type on the development of potential fire behavior over time for 48 km$^2$ of forest in Colorado, USA. Our findings can inform the placement of fuel treatments in similar forests to maximize their effectiveness and longevity. *Background and Objectives:* Efficient placement of fuel treatments is essential to maximize the impact of limited resources for fuels management. We investigated how the placement of treatments with respect to forest cover type affected the rate of spread, size, and prevalence of different fire types for simulated wildfires for 50 years after treatment. *Materials and Methods:* We generated an analysis landscape consisting of two cover types: stands on southerly aspects had low rates of tree growth and regeneration compared to stands on northerly aspects. We then simulated 1) thinning treatments across 20% of the landscape, with treatments exclusively located on either southerly ('south treatment') or northerly ('north treatment') aspects; 2) subsequent tree growth and regeneration; and 3) wildfires at 10-year intervals. Finally, we used metrics of fuel hazard and potential fire behavior to understand the interplay between stand-level fuel dynamics and related impacts to potential fire behavior across the broader landscape. *Results:* Although post-treatment metrics of stand-level fuel hazard were similar among treatment scenarios, only the south treatment reduced rates of fire spread and fire size relative to no treatment. Most differences in modeled fire behavior between treatment scenarios disappeared after two decades, despite persistently greater rates of stand-level fuel hazard development post-treatment for the north treatment. For all scenarios, the overall trajectory was of shrinking fires and less crown fire behavior over time, owing to crown recession in untreated stands. *Conclusions:* Systematic differences among cover types, such as those in our study area, have the potential to influence fuel treatment outcomes. However, complex interactions between treatment effects, topography, and vegetation structure and dynamics warrant additional study.

**Keywords:** forest inventory and analysis; forest vegetation simulator fire and fuels extension; wildland-urban interface fire dynamics simulator-level set; fuel treatment effectiveness; fuel treatment longevity

## 1. Introduction

Over a century of past land management practices including fire exclusion, timber harvesting and grazing have contributed to greater structural homogeneity in many dry, western USA conifer forests at stand and landscape scales [1–5]. These changes in forest structure, alongside increasing drought

and higher temperatures associated with climate change, have resulted in increased susceptibility to crown fire initiation and spread, along with more frequent, uncharacteristically large, high severity wildfires within western USA landscapes [6–8]. Similar trends have been observed in other parts of the world [9–12]. In response to these changes, forest managers often design and implement treatments to moderate fire behavior by manipulating the amount and arrangement of fuels in forest stands. This generally consists of fuel reduction through treatments that often involve thinning and / or prescribed fire to achieve a combination of objectives that typically include reduced canopy fuel connectivity, removal of ladder fuel, and reduced surface fuel load [13,14]. Many of these treatments occur in dry, mixed-conifer forests which are characterized by spatial patterns in forest cover types that are related to topography.

In the dry, mixed-conifer forests of the western USA, topographic complexity directly and indirectly influences several key ecological processes related to fuel treatment effectiveness, including species composition, productivity, and tree regeneration rates [4]. At the scale of a typical project area in this region (dozens of km$^2$), this often manifests as obvious spatial associations between cover types and topographic characteristics related to site dryness, such as aspect or slope position [15], late seral, fire-intolerant tree species may be more prevalent in the wettest locations, which have historically supported denser stands of trees that were more likely to support crown fire initiation and spread compared to more open forests in drier locations [4,16,17]. Thus, the immediate reduction in fire hazard from fuel treatments may be greatest in stands in relatively wet locations such as those on northerly aspects in these forests, where pretreatment fuel loading, continuity, and initial hazard are likely greater [18], and tree species' tolerance to fire is potentially lower. However, a growing body of work on long-term fuel treatment effectiveness has suggested that, in the absence of maintenance, treatment effects decline as a function of time since treatment, and that the rate of this decline is positively correlated with forest productivity and regeneration rates [19]. This implies the same ecological processes that produce elevated fuel hazards in the wettest locations may also work to shorten the lifespan of treatment effects.

Rates of fuel hazard development are potentially distinguished between cover types in this region by systematic differences in tree regeneration rates, species composition, and productivity. The influence of tree regeneration rates may be particularly strong [20]. In the Colorado Front Range, regeneration rates have been shown to be 200%–400% greater for Douglas-fir (*Pseudotsuga menziesii var. glauca* [Mirb.] Franco) dominated stands on northerly aspects compared to ponderosa pine (*Pinus ponderosa* ex. Lawson) dominated stands on adjacent southerly aspects [21]. This greater prevalence of late seral, shade tolerant tree species allows for higher densities of juvenile trees to become established underneath forest canopies, as well as for retention of longer crowns as stands develop compared to stands with proportionally more shade intolerant tree species [22]. Reduced moisture stress among trees in cover types that occur in the wettest locations typically results in greater rates of tree growth and higher stand densities than in drier locations [23]. Collectively, these factors may result in greater susceptibility to crown fire initiation and spread for Douglas-fir cover types compared to adjacent ponderosa pine types in this region. Furthermore, these findings suggest that fuel treatment effectiveness is likely to be more persistent in the ponderosa pine type, where vegetation growth and tree regeneration rates are slow compared those in the Douglas-fir type [24,25]. Thus, managers seeking to utilize fuel treatments to reduce negative impacts of wildland fires may be faced with a choice between targeting Douglas-fir-dominated stands to provide short-term benefits vs. targeting ponderosa pine-dominated stands to provide long-term benefits.

In this study, we set out to understand specifically how treatment placement within different forest cover types impacted the initial effectiveness and longevity of fuel treatments to modify potential fire behavior across a 48 km$^2$ forested landscape. This was accomplished by linking combined forest inventory observations, forest growth and yield modeling and fire simulations. We had two overarching expectations: first, preferentially treating stands in cover types associated with wetter, northerly aspects with higher surface and canopy fuel loads rather than stands in cover types associated with drier, southerly aspects would be a more effective means of modifying potential fire behavior across the study

　　　　　　　　　　　　　　　　　　　　　　　　　　　　　　　　　　　

area than vice versa. Second, any changes to potential fire behavior following treatment would be more persistent when treatments were located in cover types typical of southerly aspects with lower rates of tree regeneration and growth compared to locating treatments in cover types typical of northerly aspects.

## 2. Materials and Methods

### 2.1. Overview

We used USA Forest Service Forest Inventory and Analysis (FIA) data [26,27] and a 10 m digital elevation model (The National Map. Available online: https://nationalmap.gov) to develop an analysis landscape consisting of drier southerly aspects dominated by ponderosa pine, with relatively low productivity, tree regeneration rates and initial fuel loading; and wetter northerly aspects dominated by interior Douglas-fir, that have relatively greater productivity, regeneration rates and initial fuel loading. Using version 1969 of the Forest Vegetation Simulator and its Fire and Fuels Extension (FVS-FFE [28,29]), we simulated hazardous fuel reduction treatments across 20% of the landscape in alternative configurations such that treatments were randomly located on either southerly or northerly aspects exclusively. Then, we modeled tree growth and regeneration in treated stands for five decades. Finally, we simulated a single fire event under each treatment scenario immediately after treatment, as well as at 10-year intervals thereafter, to investigate the longevity of treatment effects under each scenario using the Wildland-urban interface Fire Dynamics Simulator subversion 9977 with level set fire spread (WFDS-LS [30]).

### 2.2. Development of Analysis Landscape

We selected an $8 \times 6$ km landscape within the Arapaho-Roosevelt National Forest, approximately 10 km east of Estes Park, Colorado, USA. for this study (Figure 1a) because aspect-driven differentiation of forest cover therein is common to mountainous terrain of many western USA forests. The dominant vegetation cover types are ponderosa pine forest, generally on southerly aspects, and Douglas-fir forest on northerly aspects (Landfire.gov; accessed 29 January 2018). Elevations range from ~1930–2830 m above sea level (Figure 1b). Slope ranges from 1% to 66%, averaging 20%. Using base tools in ArcGIS 10.3 (ESRI, Redlands, CA, USA), we first classified a digital elevation model into polygons covering northerly (270–90°) and southerly (90–270°) aspects. We then superimposed a 400 m $\times$ 400 m grid to subdivide aspect-based polygons into stands. This resulted in 1187 stands which ranged in size from 1–16 ha, averaging 4 ha (Figure 1c).

We drew from the FIA database to generate a pool of initial vegetation conditions for populating our analysis landscape (Table 1). We used only FIA plots with no recent disturbance and estimated stand ages of at least 80 years. All plots were located within the Arapaho-Roosevelt National Forest as well as within the elevation range of our study area. Plots were classified as either southerly and ponderosa pine-dominated, or northerly and Douglas-fir-dominated, where the dominant tree species accounted for ≥50% of plot basal area and stem density. While we recognize that productivity and cover type are not perfectly correlated with aspect, we used the aspect as a convenient means of generating a spatial pattern of contrasting cover types that was not excessively artificial. To ensure there were productivity differences between the forest types, we first estimated crown biomass production for each plot based on initial stand conditions using FVS-FFE. Next, we overlaid frequency distributions of biomass production for the two forest types to identify a natural break within the bimodal structure at 0.0168 kg m$^{-2}$ year$^{-1}$ and retained only northerly Douglas-fir-dominated plots producing >0.0168 kg m$^{-2}$ year$^{-1}$ (18 plots) and southerly ponderosa pine-dominated plots producing <0.0168 kg m$^{-2}$ year$^{-1}$ (26 plots). Data from the selected FIA plots were scaled to a per-ha basis and randomly assigned to either northerly or southerly facing aspects according to their species composition. Filtering the FIA data by stand age, species composition, aspect and productivity allowed us to construct an analysis landscape of undisturbed forest, where stands on northerly and southerly aspects contrasted strongly in species composition and productivity.

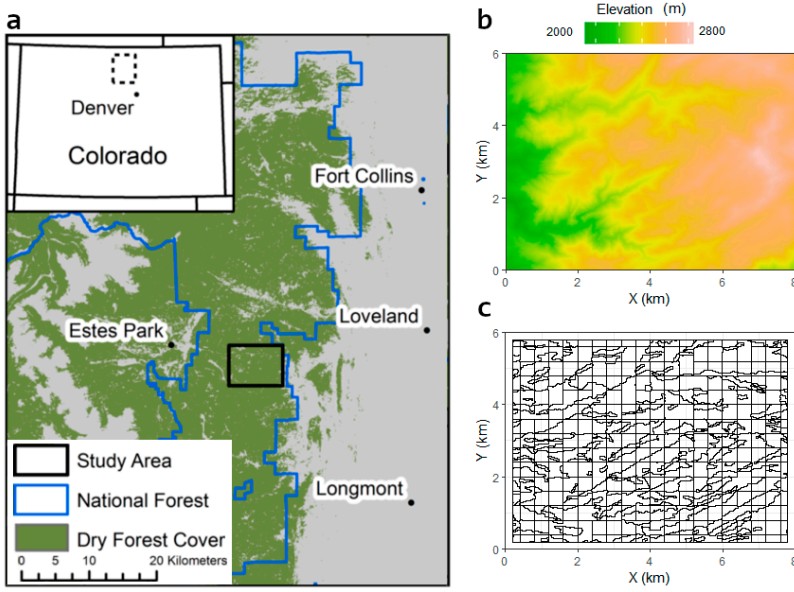

**Figure 1.** Location of our (**a**) 4800 ha study area in context of the Arapaho-Roosevelt National Forest and similar, dry forest cover types of the northern Front Range, CO, along with the (**b**) elevation and (**c**) the stand delineation of the study area. Note that our simulated landscape in (**b**,**c**) is flipped on the *x*-axis such that *x* increases from east to west.

**Table 1.** Summary of structural attributes for stands from the FIA database before and after simulated treatments. Columns are the mean (standard deviation) of site index, base age 100 ($SI_{100}$); trees per hectare (TPH); basal area per hectare (BA); quadratic mean diameter (QMD); canopy base height (CBH); and canopy bulk density (CBD).

| Aspect | Status | $SI_{100}$ (m) | TPH | BA (m$^2$ ha$^{-1}$) | QMD (cm) | CBH (m) | CBD (kg m$^{-3}$) |
|---|---|---|---|---|---|---|---|
| **Northerly** | Untreated (*n* = 18) | 15.7 (2.9) | 730 (398) | 36.5 (17.7) | 26.3 (4.5) | 1.90 (1.01) | 0.174 (0.060) |
|  | Treated (*n* = 18) | 15.7 (2.9) | 218 (73) | 11.6 (9.1) | 26.9 (4.0) | 3.25 (0.94) | 0.058 (0.015) |
| **Southerly** | Untreated (*n* = 12) | 13.9 (1.9) | 755 (549) | 24.1 (10.1) | 22.7 (5.9) | 2.31 (0.91) | 0.092 (0.047) |
|  | Treated (*n* = 12) | 13.9 (1.9) | 287 (114) | 11.9 (0.3) | 24.2 (4.8) | 3.81 (1.27) | 0.042 (0.008) |
|  | Untreatable (*n* = 14) | 14.3 (2.0) | 280 (171) | 12.6 (3.7) | 25.9 (5.7) | 3.86 (2.46) | 0.037 (0.013) |

## 2.3. Simulation of Fuel Reduction Treatments and Stand Development

We used FVS-FFE to simulate hazardous fuel reduction treatments as well as subsequent stand development. FVS-FFE is an individual-tree-based growth and yield model that predicts tree growth and fuels dynamics on a decadal time cycle using a suite of equations conditioned on species, site index [31] and regional FVS variant, among other variables [28]. In this study, site index values were imported from the FIA database and other growth parameters were left to default values.

Our simulated hazardous fuel reduction treatments were developed to emulate current silvicultural practices on the Colorado Front Range [32–34]. Treatments occurred in year 0 of the simulation and consisted of low thinning [35] of trees with a diameter at breast height <10 cm to a target quadratic mean diameter of 20–30 cm, combined with free thinning across all diameters >10 cm to a residual basal area target of 11.5 m$^2$ ha$^{-1}$ (Table 1). Fuel treatments did not depend on forest type with one exception: simulated thinning on southerly aspects targeted early seral, shade tolerant species for removal at 90% harvesting efficiency, before thinning later seral, shade intolerant species at 80% harvest efficiency; while the inverse was true for stands on northerly aspects.

Three treatment scenarios were developed to permit assessment of the impact of treatment placement with respect to patterns of forest cover type (Figure 2). In two of the scenarios, treatments

were simulated in stands comprising 20% of the landscape, while in one scenario no stands were treated ('no treatment'). In the two treatment scenarios, stands were randomly selected for treatment without regard for pretreatment stand-level fuel hazard (i.e., canopy base height [CBH], canopy bulk density [CBD]), such that all treated stands were either located in the stands with Douglas-fir cover type on northerly aspects ('north treatment'), or in the stands with ponderosa pine cover type on southerly aspects ('south treatment'). While assigning treatments to stands randomly was not a realistic representation of management practices, it allowed us to distribute treatments evenly within each of the two cover types. Data from 14 of the 26 FIA plots used to populate southerly stands in this study indicated pretreatment fuel hazards were below the treatment target, so stands populated using these data were excluded from selection in the south treatment scenario (Table 1). Our simulated treatments would have had no effect in these stands (no trees would have been cut in FVS-FFE), so including them would have reduced our capacity to assess treatment effects.

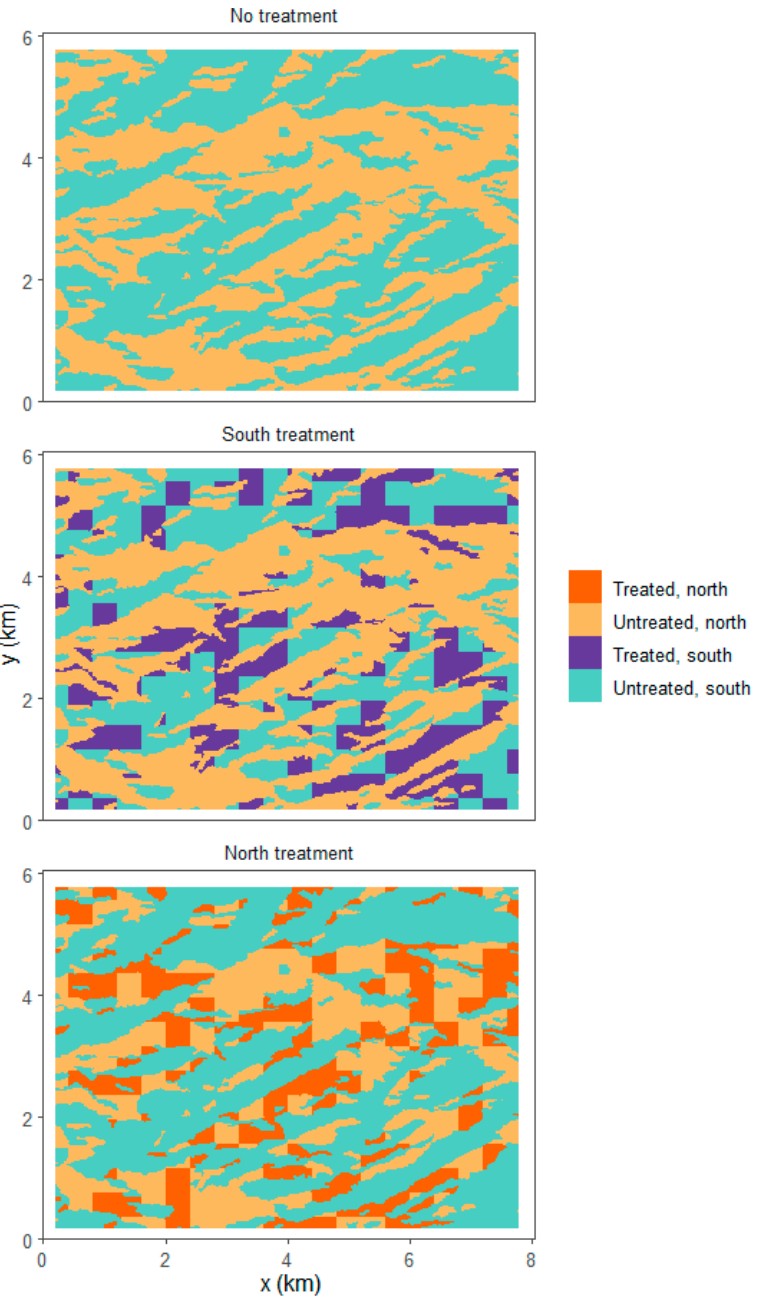

**Figure 2.** Aspect delineation and location of treatments for each of the three scenarios.

We used recent investigations of conifer seedling establishment following fuel treatments in and near the study area as guides for specifying the quantity and composition of post-treatment regeneration for each cover type [21,36]. Stands on southerly aspects received 432 ponderosa pine and 123 Douglas-fir seedlings per hectare, while stands on northerly aspects received 385 ponderosa pine and 553 Douglas-fir seedlings per hectare. Our premise was that regeneration would be precipitated by treatments [21], so all seedlings were input into the simulation as 0.30 m tall trees in the year following treatment, with 0% mortality for 10 years, at which point the mortality logic in FVS-FFE controlled seedling survival. We did not simulate tree regeneration in untreated stands, assuming that there was no available growing space for tree regeneration in the putatively undisturbed stands we selected from the FIA database.

FVS-FFE incorporates predefined fuel loadings to represent surface fuels [29]. Surface fuel models in FVS-FFE change over time depending on stand conditions, which can strongly influence model predictions of fire behavior. These changes can obscure the effects of tree regeneration and growth. As this work was specifically focused on tree regeneration and growth impacts to potential fire behavior, surface fuels for southerly aspects dominated by ponderosa pine were simulated as a fuel model 9 (medium hazard timber litter) with a dead fuel moisture of 6%, while surface fuels on the Douglas-fir dominated northerly aspects were simulated as a fuel model 10 (high hazard timber litter) with dead fuel moisture of 9% and a live fuel moisture of 30% [37]. Surface fuel model assignments were then held constant throughout all simulations. Crown fuels were modeled to be dynamic, with stand-specific parameters: FVS-FFE estimates initial crown fuel quantities (e.g., CBH, CBD) from inventory data using allometric models and then uses parameters that are species- and site-specific to model subsequent change [28].

*2.4. Wildfire Simulation*

To simulate the effect of treatments on fire spread across the analysis landscape, we used a level-set-based fire propagation model in WFDS [30,38,39]. WFDS is based on the Fire Dynamics Simulator developed at the National Institute of Standards and Technology [40,41] and allows the user to select among a suite of modeling approaches ranging in complexity from three-dimensional physics-based computational fluid dynamics models [39] to simplified, empirically-driven two-dimensional level set models [30]. Although the suit of approaches available in WFDS provides the user considerable flexibility to study fire dynamics, the physics-based approaches are computationally demanding and thus are currently limited in their ability to address questions at large spatial and temporal scales. Alternatively, WFDS-LS approaches can estimate two-dimensional fire front propagation across large landscapes faster than real time by utilizing existing empirically-derived models of fire spread and shape [30]. WFDS currently allows users to select among several different empirically-derived wildland fire behavior spread models. For this project, the fire type (e.g., surface, active and passive) and the resulting fire rate of spread were estimated by linking Rothermel's [42] surface fire spread model with the crown fire initiation and spread models of Van Wagner [43] and Rothermel [44] as implemented by Scott and Reinhardt [45]. WFDS-LS uses a flux-limiting scheme to provide smooth solutions to the elliptical fire spread equations [30]. We employed the minmod flux limiter as described by Rehm and McDermott [46] for all simulations. All technical details of the elliptical fire spread model and solution procedures for WFDS-LS are provided in Bova et al. [30].

All WFDS-LS simulations were conducted using a domain that measured 8000 m × 6000 m × 920 m that was discretized as a mesh comprised of 20 × 20 m cells. Like other fire growth models [47–49], WFDS-LS requires the user to provide gridded ASCII data that describe the topography, fuels, fuel moisture, and wind velocity and direction. In our simulations, we used the default relative humidity (40%) and temperature (20 °C) and pressure (101325 Pascal). It is important however to recognize that these values are independent of the fuel moistures prescribed in WFDS. Slope and aspect were described in WFDS-LS by first extracting the slope and aspect from the 10 m digital elevation model (The National Map. Available online: https://nationalmap.gov) using base tools in ArcGIS 10.3 (ESRI,

Redlands, CA, USA) and then reclassifying this data using a majority filter to a 20 m × 20 m resolution. Surface fuels were simulated as described in Section 2.3 above. Canopy fuel properties (i.e., CBH and CBD) were populated in each time step based on the FFE-FVS predictions as described in Section 2.3, with a foliar moisture content of 100% as suggested by Agee et al. [50]. To generate the wind velocity and direction datasets, we developed a "wind-only' simulation based on our 20 × 20 m gridded topographic data. Wind flow was entered into the simulation along the x = 0 plane following an atmospheric power-law profile with a velocity of 12 m s$^{-1}$ at 10 m above ground. We allowed the wind to flow across the domain for 0.8 h, at which time we extracted gridded wind velocity data 20 m above the terrain. These open wind speeds were then adjusted to midflame wind speeds following Andrews [51]. All other values in the simulation were set at the defaults for WFDS.

Metrics from our fire spread simulations included fire size, mean rate of spread and the proportion of area burned with surface, passive and active fire behavior. The fastest-spreading of the scenarios reached the end of the simulation domain in 7.5 h. We extracted measures of potential fire behavior from all scenarios at 7.5 h of simulated time to facilitate comparisons across scenarios. We used the spatial extent of burned grid cells to characterize fire size and the average rate of fire spread. Following typology used by the Canadian Forest Fire Behavior Prediction System [52], we classified fire behavior for grid cells as 'surface', 'active crown', or 'passive crown' according to whether crown fraction burned was <10%, >90%, or between breakpoints, respectively. Then, we tabulated the relative frequency of each fire type within each simulated fire footprint.

## 3. Results

### 3.1. Immediate Effects of Treatments

Simulated treatments reduced stand density, basal area and CBD, while increasing CBH relative to pretreatment stand conditions on both southerly and northerly aspects (Table 1). Pre- to post-treatment changes in stand structure metrics were relatively greater for stands on northerly aspects compared to stands on southerly aspects apart from QMD, which increased by 2% and 6%, respectively. Stand density, basal area and CBD were reduced pre- to post-treatment by approximately 70% for stands located on northerly aspects and 50%–60% for stands on southerly aspects. Post-treatment stand-level CBHs increased by 71% and 64% for treatments located on northerly and southerly aspects, respectively. Overall, pre- to post-treatment changes to forest structure as well as associated changes in midflame wind speed were similar for stands on northerly and southerly aspects (Table 1, Figure 3). The south treatment scenario resulted in a ~25% reduction in both simulated fire size and mean rate of spread immediately following treatment relative to the no treatment scenario, while the north treatment scenario resulted in little to no difference in simulated fire size (Figure 4a, Figure 5). Relative to the no treatment scenario, the proportion of both active and surface fire behavior types were decreased in favor of increased passive crown fire in the north treatment scenario. In the south treatment scenario, the proportion of active and passive crown fire was reduced, while surface fire increased proportionally (Figure 6).

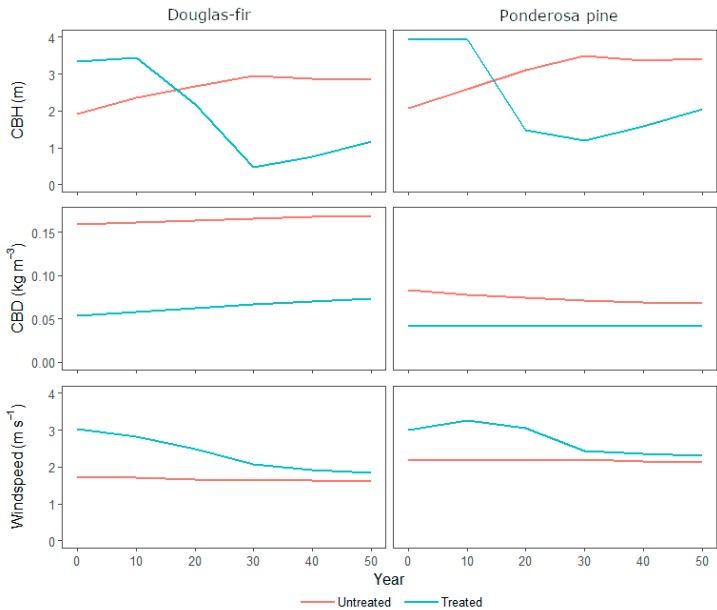

**Figure 3.** Average CBH (canopy base height), CBD (canopy bulk density) and midflame wind speed over the simulated time periods, by cover type, between treated and untreated stands. The plotted lines are averages of stand-level conditions from across all three scenarios.

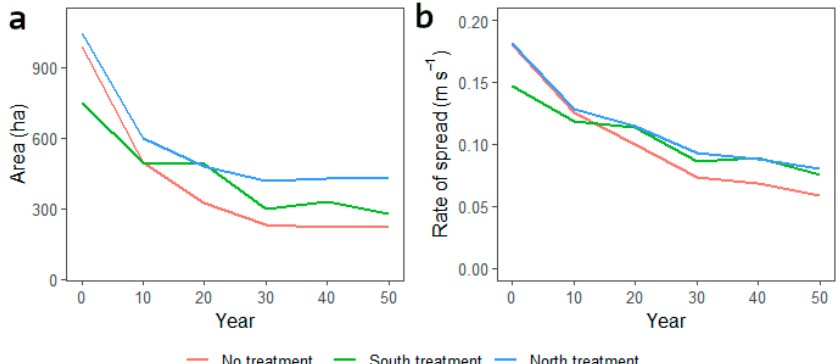

**Figure 4.** Fire size (**a**) and mean rate of spread (**b**) after 7.5 h of fire spread for all treatment cenarios.

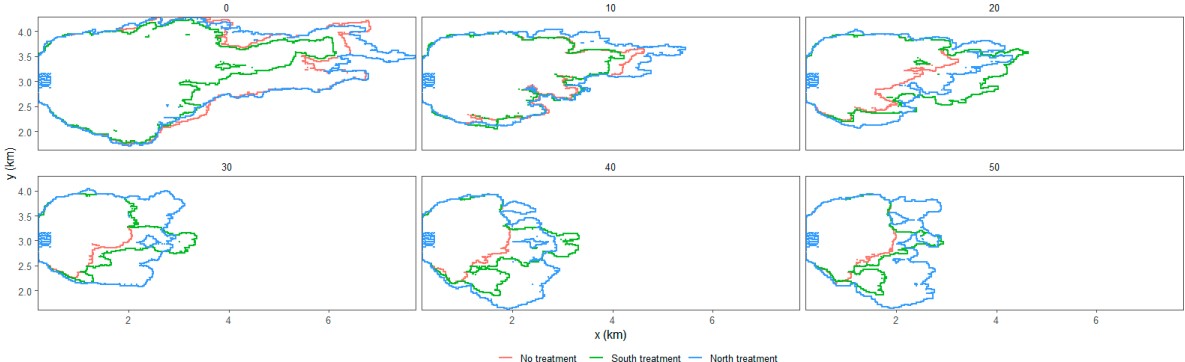

**Figure 5.** Footprints of fire areas after 7.5 h of fire spread for all treatment scenarios, after each decade of simulated forest growth.

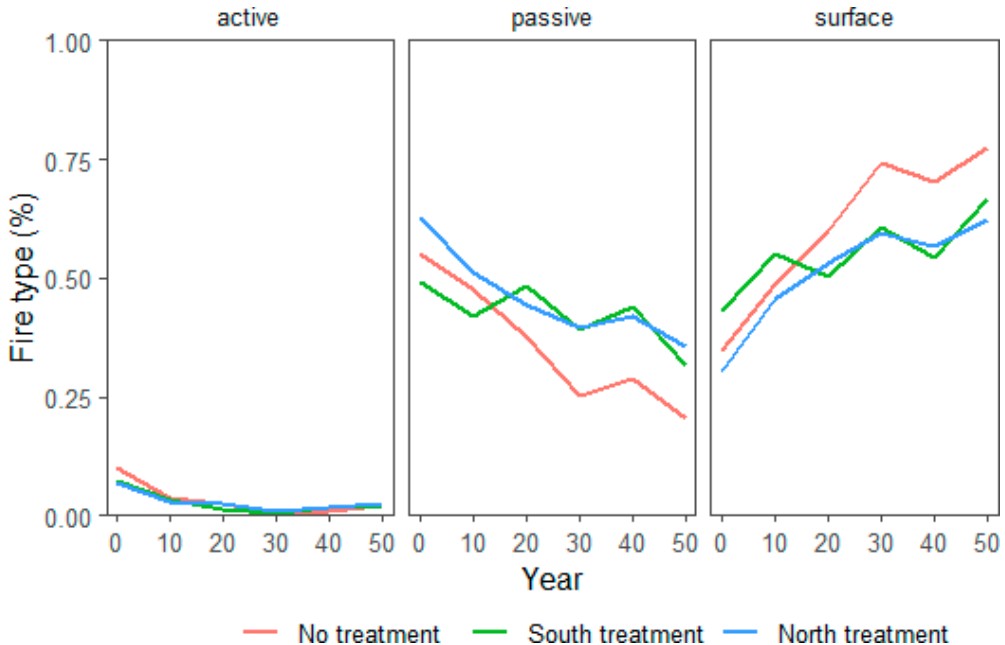

**Figure 6.** Mean percentage of fire area burned as active crown fire, passive crown fire and surface fire across all scenarios at each time step in the simulation, broken out by aspect and treatment status.

*3.2. Effects of Treatments over Time*

In untreated stands in all three of the scenarios, CBH increased from approximately 2–3 m over the first 30 years of the simulations, at which time CBH stabilized over the remaining 20 years (Figure 3). In the same stands, CBD either increased slightly or decreased through time (Figure 3). Treated stands on both southerly and northerly aspects had greater CBHs than untreated stands for the decade following treatment. After the first decade, CBH rapidly declined reaching a low point of ~1 m 30 years after treatment, followed by steady increases in CBH for the remainder of the simulation. As in untreated stands, CBD was stable through time for treated stands on southerly aspects In contrast, CBD increased by ~10% during each decade of the simulation for stands on northerly aspects. Midflame wind speeds remained elevated in treated stands relative to untreated stands throughout the simulation period (Figure 3).

The south treatment scenario resulted in decreased fire size and mean rate of spread for the first decade compared to the other scenarios. The north treatment scenario resulted in slightly greater fire size, but approximately the same mean rate of fire spread compared to the no treatment scenario over the same period (Figure 4a,b, Figure 5). Following the first decade, both treatment scenarios resulted in larger fires and greater rates of spread than the no treatment scenario. Relative to the north treatment scenario, the south treatment scenario tended to result in similar rates of spread, but ultimately smaller fire sizes (Figure 4a,b, Figure 5). For all scenarios, the proportion of passive crown fire decreased through time while the proportion of surface fire increased (Figure 6). The south treatment scenario resulted in less passive crown fire behavior relative to surface fire behavior compared to the north treatment scenario, but only for 20 years following treatment. After 20 years, both treatment scenarios resulted in more passive crown fire behavior relative to surface fire behavior compared to the no treatment scenario (Figure 6).

## 4. Discussion

In our study, we expected that initial reductions in fuel hazard and associated changes in fire behavior (i.e., fire type, fire size and spread rate) would be most strongly realized in the north treatment scenario where the denser Douglas-fir-dominated forests would yield crown fires of greater severity and faster fire growth pretreatment than the sparser ponderosa pine-dominated stands on southerly

aspects. Comparison of stand structure metrics clearly indicate that, prior to treatment, stands in the Douglas-fir type on northerly aspects did have greater CBD and lower CBH on average (Table 1). Furthermore, treatment impacts were asymmetric, reducing CBD more so for these stands. While this finding partially supports our expectation, we did not see concomitant reductions in fire size nor rate of spread in the north treatment scenario relative to the south treatment scenario (Figure 4a). Our results indicated an immediate reduction in the proportion of the analysis landscape predicted to experience active crown fire regardless of where treatments were initially placed (Figure 6). These results lend support to previous work that suggests that mechanical treatments similar to those we simulated are effective at reducing the potential for high severity fire [53]. However, the north treatment scenario also resulted in a proportional increase in passive crown fire, mirrored by a decrease in surface fires (Figure 6). This increase in passive crown fire behavior in the north treatment scenario effectively canceled out the decrease in active fire behavior, resulting in similar rates of fire spread and fire size to the no treatment scenario. Treatments in the Douglas-fir type were likely less effective at reducing active and passive crown fire as a result of increased midflame wind speeds associated with overstory canopy reduction (Figure 3), combined with greater surface fuel loads (fuel model 10 [37]) in these stands. Several previous studies have also demonstrated that increased midflame wind speeds following the removal of overstory trees can decrease resistance to passive crown fire as resistance to active crown fire increases [53–56]. In contrast to our original expectation, we found that the south treatment scenario, not the north treatment scenario, resulted in the largest decreases in the proportion of active and passive crown fires, mean rate of spread and fire size.

As we anticipated, CBD recovered at a much faster rate in Douglas-fir-dominated stands on northerly aspects. Even after 50 years, however, mean CBD was far less than pretreatment levels, indicating long-lived resistance to active crown fire (Figure 3). These findings are consistent with recent work that also showed long-lived reduction in CBD following treatments such as those simulated here, translating to prolonged reductions in the potential for active crown fire [20]. Treated stands in the Douglas-fir cover type did experience an ~10% increase in CBD per decade compared to only a slight increase over the simulation period for ponderosa pine-dominated stands, which we attribute to the systematic differences in composition, productivity and tree regeneration rates between cover types in this study. These systematic differences are also evident in somewhat greater average CBD and CBH, and slower average windspeed for untreated stands in the Douglas-fir type (Figure 3). While CBD reductions were persistent, initial increases in CBH lasted for only a single decade following treatment regardless of cover type. After the first decade, we found CBH rapidly decreased over a 20-year period as post-treatment regeneration grew sufficiently tall to be incorporated into CBH estimates for stands (Figure 3). CBH increased over the final 20 years of the simulation, likely reflecting crown recession at increasing stand densities. Although the treatment scenarios resulted in similar trajectories of CBH over time, conditions were generally somewhat more hazardous in treated Douglas-fir-dominated stands for two reasons. First, these stands reached a much lower minimum CBH over the simulation period (Figure 3). This is likely an effect of the amount and composition of tree regeneration added to these stands after treatment, which was based on observations from similar stands in and around the study area [21,36]. Stands in the Douglas-fir type had more seedlings and a larger proportion of these were shade-tolerant Douglas-fir, which maintains longer crowns at a given level of stand density compared to ponderosa pine [22]. In addition to lower minimum CBHs, treated Douglas-fir-dominated stands also showed a slower rate of CBH increase relative to ponderosa pine-dominated stands on southerly aspects (0.34 vs. 0.43 meters per decade, respectively (Figure 3)). Ultimately, the combination of lower minimum CBH, slower increase in CBH and more rapid recovery of CBD suggests that higher rates of tree growth and regeneration among stands in the Douglas-fir type will ultimately shorten the duration of stand-level treatment effectiveness in these locations.

Although there were clear differences in CBD and CBH dynamics between treated stands in the two cover types, differences among treatment scenarios in fire spread across the analysis landscape were less pronounced. Our results generally indicate that the south treatment scenario resulted in

desirable modifications to potential fire behavior for approximately two decades relative to the north treatment scenario. Following this period, there were no discernable differences among cover types in the mean rate of spread or proportional distribution of fire types among the scenarios that involved treatment (Figure 5a,b and Figure 6). This stabilization in treatment effects among scenarios was largely driven by tree regeneration and the dynamics of forest ingrowth that homogenize stand structure in the absence of further disturbance. Fire size, however, was smaller in the south treatment scenario than in the north treatment scenario (except for 20 years post-treatment, Figure 4a). Visual assessment of the simulated fire perimeters at 7.5 h suggests that the decreased fire growth that resulted in smaller fires in the south treatment scenario was associated with differences in the spatial pattern of fire perimeters relative to the other scenarios (Figure 5). These differences may reflect localized, terrain-related variations in fire spread rate and direction [57]. Terrain directly influences components of the fire environment like midflame wind speed and also indirectly influences fuels via its effect on characteristic composition and structure of forest cover types, which develop at variable rates due to variation in forest dynamics [58]. These complex interactions underlie the development of potential fire behavior over time in forests [59]. Disentangling these interactions and how they manifest in different forest types is critical to developing our understanding and ability to design fuel treatments capable of influencing fire spread across landscapes comprised of complex topography. To account for the high degree of entanglement between topography, fire weather and patterns of vegetation structure and dynamics, future studies will likely need to make use of numerical experiments where the various factors are highly controlled, and interactions can be investigated using a methodical approach [14,57].

## 5. Assumptions and Limitations

While simulation experiments such as the ones performed here allow researchers to investigate the effects of fuel treatments in ways that would be virtually impossible otherwise, results must be interpreted in light of assumptions made by the modeler. Furthermore, the scope of inference of any study is limited by the amount of data that is analyzed. Our data consisted of only one set of simulations for each treatment scenario, which precluded formal statistical testing and limited our potential scope of inference. Several key assumptions in our study design pertaining to fuel dynamics as well as to treatment design and placement are discussed in detail below.

Our results indicate that CBH dynamics were driven by crown recession and establishment and growth of regenerating trees. We used the default crown recession model in FFE-FVS, which resulted in increasing CBH through time in all untreated stands. Because most of the simulated landscape was untreated, this ultimately resulted in all simulations showing a decrease in fire size, mean rate of spread, and active and passive crown fire proportions through time (Figure 5a,b and Figure 6). These findings are consistent with the outcomes of similar studies conducted in dry forest types in California and contribute to a growing body of literature that suggests a better understanding of crown recession is critical for improving our ability to predict the longevity of fuel treatment effects [20,60,61]. These studies also highlight the importance of accounting for stand dynamics in untreated areas when assessing the longevity of treatment effects for project areas that include many stands.

This study made simplifying assumptions regarding the initial conditions and dynamics of fuels. We created an analysis landscape that was undisturbed and consisted solely of two contrasting cover types whose spatial distribution was defined solely by aspect. The inclusion of other forest types, of the effects forest health agents such as bark beetles that stress or kill trees outright, or of seasonal changes in tree chemistry that affect flammability would have added realism and undoubtedly affected our results, but also would likely have obscured the basic effects of composition, productivity and regeneration that we set out to understand. Cover types are not distributed spatially with perfect fidelity to aspect, as in our analysis landscape, but aspect was a convenient means of creating a reasonably realistic pattern of vegetation types for subsequent fire modeling. Similarly, our assumption that surface fuels were invariable within cover types, unchanging over time and, unaffected by treatments is clearly an oversimplification of the real world. Surface fuel characteristics vary considerably among stands in

their amount, moisture content and structure, and this variation is likely an influential factor dictating potential fire behavior. Our concern that this variation would obscure the specific relationships of interest to us lies behind our decision to use a common set of surface fuels for stands in each cover type. Furthermore, there is little work within the study area on surface fuel dynamics following treatment, and an even weaker empirical foundation for simulating change over time in the standard fire behavior fuel models that we used in this study [54,60,62]. Further research is required to meaningfully model changes in surface fuel over time [54].

In addition to assumptions regarding fuel dynamics, our study also made several important assumptions regarding fuel treatment design and placement. We did not simulate prescribed fire or any other means of surface fuel reduction following treatment, nor did we add activity fuels. Thus, our treatments probably best represent thinning followed by whole-tree yarding [63]. However, prior work has demonstrated that burning, in concert with thinning, can eliminate the increased potential for crown fire initiation we observed in our study [64]. Furthermore, prescribed fires can extend the longevity of fuel treatment effects by decreasing the density of regenerating trees [60,65]. In addition, our treatments produced similar overstory structures across the analysis landscape with regard for neither cover type nor pretreatment stand composition and structure. Future efforts should not only incorporate prescribed fire, but also place treatments strategically to maximize fire hazard reduction [66].

## 6. Conclusions

We examined the effects of variation in forest composition, productivity and tree regeneration rates on fire spread following installation of fuel treatments. Our results indicate that placing treatments in drier ponderosa pine-dominated stands on southerly aspects was initially more effective than placing treatments in wetter Douglas-fir-dominated stands on northerly aspects at reducing fire spread rate, size, and torching behavior in the Colorado dry, mixed-conifer forest we studied. This disparity in treatment effects lasted approximately 20 years following treatment, at which point the scenarios displayed similar fire behavior, although fire sizes generally remained smaller, and stand-level fuel hazard (CBD, CBH) continued to develop more slowly in ponderosa pine-dominated stands. Ultimately, we conclude that the placement of treatments with respect to systematic variation in composition, productivity and tree regeneration rates among cover types warrants consideration when designating areas for treatment in forests characterized by strong contrasts in these factors among cover types. However, locating treatments such that they target stands with the greatest fuel hazards, and creating treatment patches of sufficient size to disrupt fire spread may be of equal or greater importance.

**Author Contributions:** Conceptualization: S.A.E. and C.M.H.; Methodology: C.M.H., W.T.T., and J.P.Z.; Software: C.M.H. and J.P.Z.; Validation: J.P.Z.; Formal analysis: W.T.T. and J.P.Z.; Investigation: S.A.E., C.M.H., W.T.T., and J.P.Z.; Resources: C.M.H.; Data curation: C.M.H.; Writing—Original draft preparation: S.A.E. and J.P.Z.; Writing—Review and editing: S.A.E., C.M.H., W.T.T., and J.P.Z.; Visualization: J.P.Z.; Supervision: C.M.H; Project administration: C.M.H.; Funding acquisition: S.A.E. and C.M.H.

**Funding:** This work was funded by the Joint Fire Science Program, project #14-1-01-18.

**Acknowledgments:** William Mell (USDA Forest Service, Pacific Northwest Research Station) for software development of WFDS-LS.

**Conflicts of Interest:** The authors declare no conflict of interest.

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
