# Peer review of "Long-Term Impacts of Fuel Treatment Placement with Respect to Forest Cover Type on Potential Fire Behavior across a Mountainous Landscape"

_forests, doi:10.3390/f10050438_

Round 1

Reviewer 1 Report

I enjoyed reading this paper, and feel that it should be published, however I do feel there are a few issues that should be addressed.

This study says it investigates potential fire effects of fuel treatment placement with respect to cover type on landscape-level treatment in the Western U.S. and can inform the design of landscape-level fuel treatments.  However, this study more accurately examined the effectiveness between randomly placing some fuel reduction treatments on North vs South facing slopes (but maybe even more accurately, randomly placing fuel Tx's in DF vs PP forest types in the front range of Colorado.)  I think this paper could be greatly improved and do what it claims by increasing the scope to sampling other study sites within the Western US, and across a more representative sample of fuel types and regimes. I also think that 8000 m X 6000 m (800m elevation gradient) is a bit short of landscape level (Landscape scale), as suggested in much of the introduction and discussion.  With this said, I realize what I just suggested may fundamentally change the scope and objective of the authors, yet this paper still seems publishable if framed as examining the importance of cover type (maybe aspect) for the the longevity and effectiveness of treatments.  I think the discussion/conclusion (and intro) probably need to narrow the implications to the types and region studied. 

The strength of this paper is the modelling approach, and this use of the methodology to examine a fuels and fire variable while holding other variables constant.  (This is mentioned on line 378).  The authors then discuss some assumptions made in the modelling, yet I think there are many more assumptions, so many that the authors might want to consider a “Assumptions and Limitations” section.  The assumptions I identified that could greatly change results:  constant live fuel moistures, and surface fuel moistures (these could vary greatly from north to south, and at different latitudes). Canopy density (and CBD by type) e.g. a ponderosa (PP) on a productive site would have a much different CBD, CBH, etc, than on a poor site at the extreme ends of its range. That variable physiology like is constant, e.g. tree chemistry has been show to change across gradients, latitudes and season, and affect flammability. That disturbances (e.g. MPB, defoliators, drought stress, etc.) are absent on the landscape.  That all PP is fuel  model 9, all DF is model 10, that dead fuel moistures would remain constant. That PP/DF represents the Western US (e.g. Lodgepole types?)  The absence of shrub understory and other species mixtures (ladder fuels), especially across an elevational gradient.  I feel the assumption of this study that aspect is the most important variable in describing forest type changes especially needs to be addressed, if this were true then species distribution maps could be made using aspect alone.  Again, I want to stress that I feel holding these variables constant is the strength of using a modelling approach and this paper, but I think some more of the assumptions need to be addressed.

Line specific comments:

Line 102-108 – the use of landscape scale should probably be defined, especially considering the study window of 8km X 6km. 

Line 113 – you may want to reference you DEM source and resolution.

Line 137 – Figure 1, your map elements overlap, e.g. your study area box bleeds into you map, as well as your scale bar.

Line 142 – why do you assume that disturbance is not part of a natural landscape?

Line 146-150 – productively and forest type depends on more than aspect.  Also, transformed aspect (e.g. SW based) could change results.

Line 171 – 180 – Land managers don’t randomly select squares in mountainous terrain for fuels treatments.  Also, why did you not include the FIA plots with low densities/stocking?  This is part of a natural landscape.   

Line 203 – you may want to describe and cite the fuel models. (Anderson, 1982 or similar)

Line 222 – you may want to define mid-mod limiter

Line 226 – “Like other landscape model”  need citation

Line 228 – Relative humidity?

Line 230 – See above line 113 on citing DEM (personally, I think if you are resampling terrain data you should resample the DEM and then create any derived terrain variables such as slope or aspect to avoid artifacts, however it probably didn’t affect you fire simulations)

Line 350 – Figure 3, I am not sure this graphic adds much, maybe if you included simulation results in the figures.  Also, I am confused, in figure 1b it looks like the landscape is highest at the west, but this graphic makes it look highest at the east.  I also could not find you wind direction of the landscape which would greatly affect fire simulations.

Line 274 – Figure 4 – you legend does not match your plot (there is a dash line in your legend)  Also, why did you not include your control (untreated) in this figure?

Line 278 – Figure 6 – All of those lines on these plots look like they could be statistically the same.  I think a significance test (e.g. time series analysis) would make your results much more compelling.  Also, a few more iterations of your simulation would help too.  E.g. You randomly select some 400mX400m treatment locations, why don’t you replicate this a few times, that way you could do a significance test. This figure also needs to be higher quality.

Line 314-316 – “. . . support to several previous studies” – You cite one, and Agee and Lolley report mixed results that were dependent on scale.

Line 340-344 – you should mention how your treatments differ from your control. This is lacking in other places in the paper too.

Line 374-378 – Really good discussion.

Paragraph at 378 – I think this is very good discussion on the modelling and studies limitations.  I just think you need to address more of the assumptions and limitations.  See above comment.

Line 394 – agreed, however there has been some work done on Dynamic surface fuels pathways post fire.

Line 414 – fire size may not be the best (or primary) metric for fuel reductions success, e.g. fire severity, proximity of WUI.

Line 423 Supplemental figure – I actually think this figure is important, you may be able to combine Fig 1 and 3, or combine this fire with figure 3.  However, you should make your X and Y axis the same scale, especially when showing geographic data.

Author Response

Note – all review comments are reproduced below, with our responses to each comment inserted in the text in italics. Where our responses are accompanied by revisions in the manuscript, we specify line numbers corresponding to the revised text in parentheses.

Reviewer 1

This study says it investigates potential fire effects of fuel treatment placement with respect to cover type on landscape-level treatment in the Western U.S. and can inform the design of landscape-level fuel treatments.  However, this study more accurately examined the effectiveness between randomly placing some fuel reduction treatments on North vs South facing slopes (but maybe even more accurately, randomly placing fuel Tx's in DF vs PP forest types in the front range of Colorado.)  I think this paper could be greatly improved and do what it claims by increasing the scope to sampling other study sites within the Western US, and across a more representative sample of fuel types and regimes. I also think that 8000 m X 6000 m (800m elevation gradient) is a bit short of landscape level (Landscape scale), as suggested in much of the introduction and discussion.  With this said, I realize what I just suggested may fundamentally change the scope and objective of the authors, yet this paper still seems publishable if framed as examining the importance of cover type (maybe aspect) for the the longevity and effectiveness of treatments.  I think the discussion/conclusion (and intro) probably need to narrow the implications to the types and region studied. 

As suggested, we have re-framed the paper to focus on the importance of cover type, instead of on landscape-level fuel treatments and fire behavior. This involved many revisions throughout the manuscript. While we retained the term ‘landscape’ to describe our study area, we adjusted our wording to refer more specifically to ‘our analysis landscape’ or similar, instead of referring generally to landscapes or ‘landscape-level’ effects / phenomena. Some examples of this: We have removed references to ‘landscape-level networks’ of fuels treatments and now simply refer to them as ‘fuel treatments’ or similar, and we similarly excised references to ‘landscape-level’ fire behavior in favor of simply ‘fire behavior’ or similar  (11-16, 23-31, 36, 83-97, 206, 225, 307-308, 316, 321, 359-362, 374-376, 387, 400-402, 428-430, 433-443, Fig. 2 caption) We also removed a paragraph from the introduction that provided background on landscape fuel treatments.

Regarding narrowing the implications to the types and regions studied, we appreciate that our findings are unlikely to be generally true for all forests, but we maintain that the implications of this work are nonetheless relevant outside of the specific types and region we studied. Specifically, our findings can inform management of forests with active fuel management programs that are characterized by systematic spatial patterns of forest cover type that are related to topography (i.e. many dry, mixed-conifer forests). We clarified the scope of the work in several places throughout the manuscript (13-14, 31-32, 50-52, 55-58, 389-391, 400-402, 434-438, 441-444). We also revised the introduction to focus more specifically on dry, mixed-conifer forests (41-42, 53, 71-74).

The strength of this paper is the modelling approach, and this use of the methodology to examine a fuels and fire variable while holding other variables constant.  (This is mentioned on line 378).  The authors then discuss some assumptions made in the modelling, yet I think there are many more assumptions, so many that the authors might want to consider a “Assumptions and Limitations” section.  The assumptions I identified that could greatly change results:  constant live fuel moistures, and surface fuel moistures (these could vary greatly from north to south, and at different latitudes). Canopy density (and CBD by type) e.g. a ponderosa (PP) on a productive site would have a much different CBD, CBH, etc, than on a poor site at the extreme ends of its range. That variable physiology like is constant, e.g. tree chemistry has been show to change across gradients, latitudes and season, and affect flammability. That disturbances (e.g. MPB, defoliators, drought stress, etc.) are absent on the landscape.  That all PP is fuel  model 9, all DF is model 10, that dead fuel moistures would remain constant. That PP/DF represents the Western US (e.g. Lodgepole types?)  The absence of shrub understory and other species mixtures (ladder fuels), especially across an elevational gradient.  I feel the assumption of this study that aspect is the most important variable in describing forest type changes especially needs to be addressed, if this were true then species distribution maps could be made using aspect alone.  Again, I want to stress that I feel holding these variables constant is the strength of using a modelling approach and this paper, but I think some more of the assumptions need to be addressed.

We added a subheading titled ‘Assumptions and Limitations’, as suggested. This section is an expansion of the final two paragraphs of the discussion section from the first version of the manuscript. We added text addressing the potential effects of holding fuel models, moisture content and tree flammability constant, as well as of creating an analysis landscape that was artificial in that it consisted of only two cover types and was wholly undisturbed prior to treatment (404-409, 413-417). We made extensive changes throughout the manuscript to re-focus on cover type rather than aspect (55-63, 67-86, 91-97, 135-137, 312-314, 323, 331-332, 335-342, 345-363, Fig. 3 caption), and also addressed simplifying assumptions related to aspect here (404-405, 409-411).

We feel two of the assumptions identified by the reviewer do not warrant inclusion in the section discussed above: First, FVS-FFE incorporates site- and stand-specific parameters to estimate CBD and CBH, so site-related variability in these factors is captured in our modeling approach. We added text clarifying this to the methods (201-204). Second, we agree the implicit assumption that ponderosa pine and Douglas-fir cover types are generally representative of western U.S. forests is tenuous. By narrowing the scope of the entire paper to focus on dry, mixed-conifer forests characterized by systematic spatial patterns of cover type related to topography, we attempted to replace this implicit assumption with an explicit statement of the scope of inference of the work (see our responses to the first comment from this reviewer above).

Line specific comments:

Line 102-108 – the use of landscape scale should probably be defined, especially considering the study window of 8km X 6km. 

We did this. See our response to the first comment from this reviewer above.

Line 113 – you may want to reference you DEM source and resolution.

We did this. (101)

Line 137 – Figure 1, your map elements overlap, e.g. your study area box bleeds into you map, as well as your scale bar.

We fixed the aesthetics of the legend box in Figure 1.

Line 142 – why do you assume that disturbance is not part of a natural landscape?

We clarified that the effects of disturbance were excluded to avoid obscuring the effect of interest in the Assumptions and Limitations section (405-409)

Line 146-150 – productively and forest type depends on more than aspect.  Also, transformed aspect (e.g. SW based) could change results.

We added a sentence stating that aspect was used primarily as a means of creating a spatial pattern of contrasting cover types that was not excessively unrealistic, not as an explanatory variable (135-137).

Line 171 – 180 – Land managers don’t randomly select squares in mountainous terrain for fuels treatments.  Also, why did you not include the FIA plots with low densities/stocking?  This is part of a natural landscape.   

We added two sentences explaining the rationale for random treatment assignment and for dropping low density stands to the methods (169-171, 173-175).

Line 203 – you may want to describe and cite the fuel models. (Anderson, 1982 or similar)

We added a description of the fuel models to the text (198-200). A citation (Anderson 1982) was provided in the first version, but we notice the numbering for references was shifted such that the review copy cited Mell et al. (2007). This error has been corrected.

Line 222 – you may want to define mid-mod limiter

We did this (220-222).

Line 226 – “Like other landscape model” need citation

We added several citations to this sentence as requested (225).

Finney, Mark A. "FARSITE: Fire Area Simulator-model development and evaluation." Res. Pap. RMRS-RP-4, Revised 2004. Ogden, UT: US Department of Agriculture, Forest Service, Rocky Mountain Research Station. 47 p. 4 (1998).

Finney, M.A. 2006. An overview of FlamMap modeling capabilities. In: Andrews, Patricia L.; Butler, Bret W., comps. 2006. Fuels Management—How to Measure Success: Conference Proceedings. 28-30 March 2006; Portland, OR. Proceedings RMRS-P-41. FortCollins,CO: U.S. Department of Agriculture, Forest Service, Rocky Mountain Research Station. 809 p.

Stratton, R. D. (2006). Guidance on spatial wildland fire analysis: models, tools, and techniques. Gen. Tech. Rep. RMRS-GTR-183. Fort Collins, CO: US Department of Agriculture, Forest Service, Rocky Mountain Research Station. 15 p., 183.

Line 228 – Relative humidity?

We added two sentences to this section clarifying our modeling procedure (227-229)

Line 230 – See above line 113 on citing DEM (personally, I think if you are resampling terrain data you should resample the DEM and then create any derived terrain variables such as slope or aspect to avoid artifacts, however it probably didn’t affect you fire simulations)

We cited the DEM here as well (231).

Line 350 – Figure 3, I am not sure this graphic adds much, maybe if you included simulation results in the figures.  Also, I am confused, in figure 1b it looks like the landscape is highest at the west, but this graphic makes it look highest at the east.  I also could not find you wind direction of the landscape which would greatly affect fire simulations.

We cut Figure 3. Wind modeling is described in 235-241. As modelled, our landscape elevation does rise with increasing values of the X axis, however the landscape rises from East to West. We added a note to the caption of Fig. 1 that the axes of panels b and c are flipped.

Line 274 – Figure 4 – you legend does not match your plot (there is a dash line in your legend)  Also, why did you not include your control (untreated) in this figure?

We fixed the dashed legend key by making the key solid. Stands from the ‘no treatment’ scenario are included in the figure. We added a sentence to the caption clarifying which data are shown.

Line 278 – Figure 6 – All of those lines on these plots look like they could be statistically the same.  I think a significance test (e.g. time series analysis) would make your results much more compelling.  Also, a few more iterations of your simulation would help too.  E.g. You randomly select some 400mX400m treatment locations, why don’t you replicate this a few times, that way you could do a significance test. This figure also needs to be higher quality.

Our data do not allow formal statistical testing. We added text to the Assumptions and Limitations section explaining how this limits the scope of inference of the work (389-391). We have included a high-quality figure in this version.

Line 314-316 – “. . . support to several previous studies” – You cite one, and Agee and Lolley report mixed results that were dependent on scale.

We tempered the language here to say our study “. . . lends support to previous work that suggests . . .” (318-319)

Line 340-344 – you should mention how your treatments differ from your control. This is lacking in other places in the paper too.

This information is displayed in Fig. 3. See our response to comment about Fig. 4 (now Fig. 3) above. We also added clarifying text to the results (284) and discussion (339-340) sections.

Line 374-378 – Really good discussion.

Thank you.

Paragraph at 378 – I think this is very good discussion on the modelling and studies limitations.  I just think you need to address more of the assumptions and limitations.  See above comment.

This is addressed in our responses to comments above.

Line 394 – agreed, however there has been some work done on Dynamic surface fuels pathways post fire.

We added a citation here (419)

Davis, B., van Wagtendonk, J., Beck, J. and van Wagtendonk, K., 2009. Modeling fuel succession. Fire Management Today. 69 (2): 18-21., pp.18-21.

Line 414 – fire size may not be the best (or primary) metric for fuel reductions success, e.g. fire severity, proximity of WUI.

We added text here to also highlight differences in torching behavior between the two treatment scenarios (437).  

Line 423 Supplemental figure – I actually think this figure is important, you may be able to combine Fig 1 and 3, or combine this fire with figure 3.  However, you should make your X and Y axis the same scale, especially when showing geographic data.

We moved this figure into the body of the manuscript in this version. It is now Figure 5. We made sure that maps of our study area, in all our figures, are on linear scales with an aspect ratio of 1:1 (so as not to distort the shape of the data). We acknowledge that Fig. S1 (now Fig. 5) has a different Y range than Figs. 1 and 2; we chose not to extend the Y range to the full extent of our simulated landscape in Fig. 5, which shows fire perimeters, because there were no data in the extended range to plot.

Reviewer 2 Report

Figures 4,5,6 have not got good optical analysis Is there a better way to give the results...

Author Response

Reviewer 2

Figures 4,5,6 have not got good optical analysis Is there a better way to give the results...

This comment from Reviewer 2 does not identify any specific issues with Figs. 4-6, so we did not make any revisions in response. We did make minor changes to Fig. 4 and its caption in this version of the manuscript in response to comments from Reviewer 1.

Reviewer 3 Report

The paper is well written and well presented. This research provides quality information in the growing field of fuel treatment research. I have only very minor comments/suggestions to make in the manuscript on pages 2, 7, and 8. These are more interpretive comments and are very easily addressed. I have placed the comments in the attached manuscript.

Author Response

Reviewer 3

The paper is well written and well presented. This research provides quality information in the growing field of fuel treatment research. I have only very minor comments/suggestions to make in the manuscript on pages 2, 7, and 8. These are more interpretive comments and are very easily addressed. I have placed the comments in the attached manuscript.

Note: we have reproduced Reviewer 3’s comments below.

Line 50—‘lower rates’ is unclear

This section was stricken following other reviewer comments

Lice 51—‘reduce fire behavior’ is unclear. Extreme fire behavior? Erratic behavior?

See below response.

Line 52 – Fire behavior is a term describing how a fire acts or responds, or behaves. I’m not sure that stating reducing behavior, or reducing for a fire acts is correct. I understand what you mean but possibly stating to reduce or minimize a type of behavior would be more accurate. I believe you are talking about extreme behavior, or at least high intensity fires. Fires can behave normally or abnormally, and I believe it is the latter you are referencing.

We changed our wording from “reducing” to “modifying” or similar throughout the text (30, 93, 362).

Line 88—Replace ‘below established’ with ‘to become established under’.

We incorporated the reviewer’s suggestion (76).

Line 219-220—Remove ‘see this section, below’. Remove comma between ‘active’ and ‘and passive’.

We did this (217).

Lines 241-242—Remove ‘see classification criteria below’

We did this (243).

Line 251—Axis should be labeled to prove better clarity

Following comments from another reviewer, we have opted to remove this figure altogether.

Round 2

Reviewer 1 Report

I think the changes made by the authors have improved the manuscript.